# The Effect of Glass Flour on the Microstructure and Properties of Fiber-Reinforced Concrete: Experimental Studies

Gabriela Rutkowska [1,*], Mariusz Żółtowski [1], Filip Chyliński [2], Yuliia Trach [1] and Elżbieta Gortych [1]

1 Institute of Civil Engineering, Warsaw University of Life Sciences, 166 Nowoursynowska Street, 02-787 Warsaw, Poland; mariusz_zoltowski@sggw.edu.pl (M.Ż.); yuliia_trach@sggw.edu.pl (Y.T.); s200205@sggw.edu.pl (E.G.)
2 Instytut Techniki Budowlanej, Filtrowa 1, 00-611 Warsaw, Poland; f.chylinski@itb.pl
* Correspondence: gabriela_rutkowska@sggw.edu.pl

**Abstract:** The introduced limits on carbon dioxide emissions by the European Union encourage experimental work on new-generation materials containing smaller amounts of clinker. Currently, silica fly ash from hard coal combustion is widely used in cement and concrete technology in Europe and Poland. Their wide application is determined mainly by the chemical and phase composition, and in particular by the activity of pozzolanic and its high fineness, like cement. The aim of this study was to assess the effect of glass flour and polypropylene fiber modifiers on the properties of concrete and its microstructure. To analyze the results, samples of reference ordinary concrete and samples with different amounts of glass flour (0–30%) and a constant number of polypropylene fibers (0.025 kg) were used. The obtained test results showed the possibility of producing ordinary concrete with the addition of glass flour. The average compressive strength for concrete containing 10% additive was set at 49.3 MPa, 51.2 MPa, and 53.1 MPa after 28, 56, and 90 days of maturation for a content of 20% of 44.6 MPa, 46.4 MPa, and 48.4 MPa, respectively, and for 30% of 41.5 MPa, 43.8 MPa, and 45.6 MPa, respectively. By modifying concrete with glass flour and polypropylene fibers, a composite resistant to negative temperatures can be obtained. Glass flour shows reactivity with the cement matrix, and in small amounts, it might cause the microstructure to seal and a slight increase in compressive strength.

**Keywords:** compressive strength; frost resistance; concrete microstructure; glass flour; fibers





## 1. Introduction

Concrete is undoubtedly the most used composite material among man-made materials and second only to water in the entire complex of materials used, without which modern construction could not function. It owes this position to well-known advantages such as high strength and durability, ease of manufacture and stacking, and low cost of production. However, this material is not perfect. According to the literature, it is susceptible to harmful biological and physical effects. According to estimates, about 3.5 billion $m^3$ of this material is consumed annually worldwide. It is exploited in increasingly difficult natural conditions, from the hot desert to the ice of the Arctic and the sea floor, from sewers to "skyscrapers". Concrete is a material with a high potential to adapt to specific operating environmental conditions. It is an ecological composite, often made of local raw materials—aggregate, cement, water, admixtures, and possibly mineral additives. It is a safe product that guarantees the stability and load-bearing capacity of a given structure but is also a sustainable, technologically advanced product [1–4].

Looking at concrete from the side of its microstructure, its durability will be shaped by the nature of the cements used, their quantity, admixture properties, proportions of binder hydration products, and water–cement (binder) coefficient [5,6].

For the development of the construction sector, one of the most important issues is striving to make concrete an ecological material that is even more environmentally friendly

so that it can meet both the current and future requirements of construction. It should fully meet current social needs and, at the same time, meet new challenges, ensuring the innovativeness of the material. It is also important that it "works" flawlessly, without the need to carry out costly repairs, so that the production of concrete is correlated with the prediction of its behavior during specific operating conditions. For a building material to guarantee economy, environmental friendliness, innovation, or durability, the structure of concrete should be improved or modified. Modification at the structure level changes the physical and mechanical properties of the material. The search for such solutions is indispensable in the case of designing the composition of a concrete mix, whose two components, cement and aggregate, contribute to anthropopressive interactions at the stage of their acquisition and production. In Poland and around the world, there are already areas where obtaining good-quality materials for cement production is a problem. Every year, the world economy needs more and more cement to produce concrete, for which, to this day, no comparable replacement has been found. The big problem is that during the production of 1 ton of cement, 0.5 to 1 ton of greenhouse gases are produced, which is 6–8% of total anthropogenic emissions, according to various data. The concrete production process has a carbon footprint estimated at about 850 kg of $CO_2$ emitted per ton of clinker needed to produce cement. The European Union's carbon dioxide emission limits (target: 55% reduction in emissions by 2030) encourage research into new generation materials containing smaller amounts of clinker [7–10].

The use of industrial waste in the composition of concrete reduces the amount of cement clinker used and natural aggregates for its production [5]. Nowadays, the materials most widely used to produce concrete are limestone and silica fly ashes and silica dust, referred to as pozzolana [11,12]. Research is being conducted on the possibility of using other components, e.g., fly ash from the thermal treatment of sewage sludge [13–16], bio-ash resulting from the combustion of wood and other plant biomass [17], and shredded glass waste [18–21]. The use of waste provides numerous environmental benefits, such as reduced landfill costs, energy savings, and reduced carbon emissions. In addition, their use can improve the microstructure and mechanical properties of mortar and concrete [22]. Existing environmental requirements are becoming more stringent [23]. In many European countries (Netherlands, Germany, Belgium), the amount of glass collected from consumers exceeds 80% of the glass produced; most of it is recycled.

Crushed glass research is focused on the use of glass as fine concrete aggregates. However, durability concerns over the alkali–silica reaction (ASR) have limited its use as a fine replacement in concrete. Studies have shown that glass behaves pozzolanically if ground finely enough, with a surface area of more than 300 $m^2$/kg [24,25]. The pozzolanic reaction produces amorphous silica in the SCM, calcium hydroxide (CH) as a byproduct of the cement reaction, and water to form additional calcium silicate hydrate. Glass effect measures such as SCM focus on mechanical and durability properties, and they show an increase in long-term compressive strength, flexural strength, resistance to ASR, and reduction in water sorptivity of concrete containing finely ground glass powder. Moreover, some studies showed that finely ground glass powder had comparable or slightly better mechanical properties at later ages than fly ash and slag, but much less than silica fume [26–28]. Despite all the aforementioned results, few studies have focused on connecting the microstructural properties of cementitious mixtures containing glass powder to the performance characteristics of glass mixtures. Federico [29] performed an extensive study on the glass powder's kinetic and performance properties. However, the effect of curing temperatures on different types of glass cullet reaction kinetics and performance has not been studied.

The aim of the study conducted by Aliabdo et al., was to determine the feasibility of using glass powder as a concrete additive. The pozzolanic activity of glass powder and the effect of this additive as a cement substitute in the range of 0% to 25% on the physical and mechanical properties of the studied composites were evaluated. The test results showed that glass flour is pozzolanic in nature and meets the limits for classes F and C according

to ASTM C 618, and the use of glass flour has a negligible effect on the setting time. The use of 10% glass powder as a substitute for cement increased the compressive strength of the mortar by about 9.0%. The use of glass powder in an amount greater than 15.0% as a substitute for cement reduces the 28-day compressive strength of concrete. In order to compensate for the decrease in the compressive strength of concrete, it is necessary to reduce the w/c ratio [30]. Experimental studies conducted by the Kishan Lal Jain and team aimed to evaluate the durability of concrete mixes, containing glass waste flour and granite powder at different levels of substitution. Glass powder (GP) in quantities of 5%, 10%, 15%, 20%, and 25% and granite powder (GrP) in quantities of 10%, 20%, 30%, 40%, and 50% were added to the mixtures as a partial addition to cement and sand, respectively. A significant improvement in the strength properties of concrete containing 15% GP and 30% GrP instead of cement and sand, respectively, was observed. The results show an improvement in water permeability and water absorption by concrete mixed with glass granite [31].

The aim of the research was to learn the impact of adding shredded glass waste and polypropylene fibers (PPFs) during the preparation of the concrete mix of ordinary concrete on its selected technical properties. The obtained results allowed us to determine the strength and frost resistance of the tested material with different contents of glass waste and the same number of polypropylene fibers. In addition, the effect of the addition of glass flour and polypropylene fibers on the microstructure of concrete was evaluated.

## 2. Materials and Methods

Procedures based on the guidelines contained in current EU regulations and standards were used to carry out the research. Figure 1 shows a conceptual diagram of the research carried out.

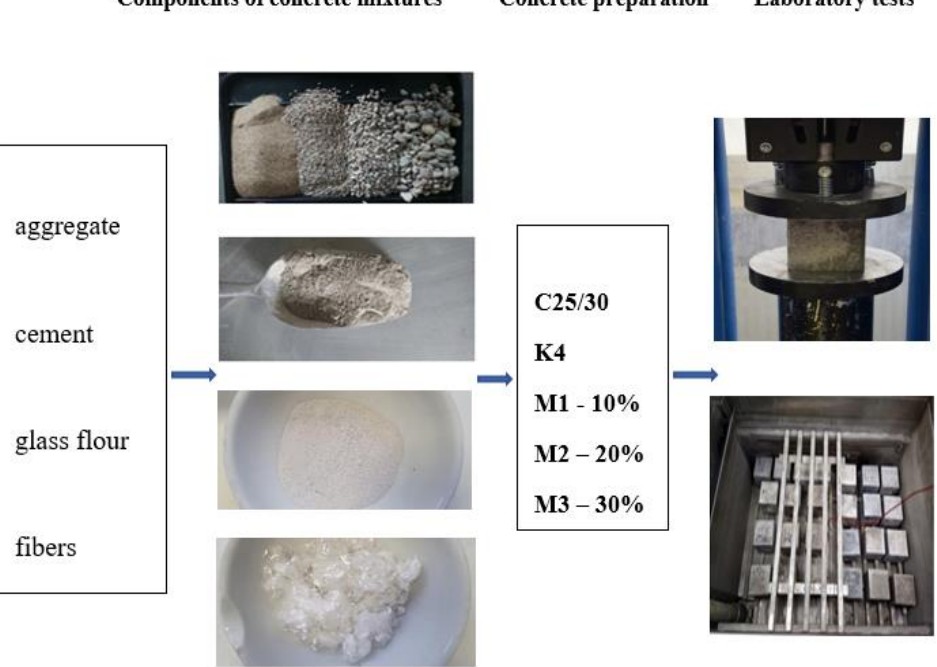

**Figure 1.** Conceptual diagram of laboratory tests performed.

To determine the effect of glass flour and polypropylene fibers on compressive strength, samples of ordinary concrete of class C25/30 with consistency K4 (semi-liquid consistency) were prepared based on EN 206 + A2:2021-08. To prepare the concrete mix, Portland cement CEM I 42.5 R from the Ożarów cement plant, in accordance with PN-EN 197-1: 2012, was used [32,33]. This cement is characterized by its high early strength. The declared

performance of Portland cement is summarized in Tables 1 and 2. The given values are average values, guaranteed by the manufacturer for 2023.

**Table 1.** Physical properties and phase composition of cement CEM I 42.5 R (data from the manufacturer—own support).

| Blaine Specific Surface Area (cm²/g) | Start of Setting Time (min) | Compressive Strength after 2 Days (MPa) | Compressive Strength after 28 Days (MPa) |
|:---:|:---:|:---:|:---:|
| 3330 | 218 | 21.0 | 49.8 |
| Share of mineral phases CEM I (% w.) | | | |
| C3S—55.54 | C2S—14.59 | C3A—8.15 | C4AF—6.85 |

**Table 2.** Chemical properties of cement CEM I 42.5R (data from the manufacturer—own support).

| Loss of Ignition (%) | Sulphate Content $SO_3$ (%) | Cl Chloride Content (%) | Alkali Content as $Na_2O_{eq}$ (%) | $SiO_2$ (%) |
|:---:|:---:|:---:|:---:|:---:|
| 3.19 | 2.96 | 0.05 | 0.76 | 20.20 |
| $Al_2O_3$ | $Fe_2O_3$ | CaO | CaOw | MgO |
| 4.41 | 2.42 | 64.36 | 1.98 | 1.98 |

In all samples, the same granulometric composition of fine aggregate, river sand of the 0–2 mm fraction selected by the sieving method according to EN 933-1:2012 and the same composition of coarse pebble aggregate fraction 4–16 mm selected by the method of successive approximations in three stages—Table 3 was assumed [34]. The bulk density of coarse and fine aggregate was determined by the pycnometric method in accordance with EN 1097-6:2002 [35]. The density for sand was 2.60 g/cm³ and for gravel, 2.65 g/cm³. For all laboratory tests, drinking water was used in accordance with EN 1008:2004 [36].

**Table 3.** Grain composition of aggregate (own version).

| Fractions | Fraction Mixing Percentage (for Sand and Gravel) | | | Particle Size (%) | |
|:---:|:---:|:---:|:---:|:---:|:---:|
| | Stage I | Stage II | Stage III | Sand | Gravel |
| 0.0–0.125 | | | | 1.93 | 0.60 |
| 0.125–0.25 | | | | 17.82 | 5.52 |
| 0.25–0.50 | | | 31 | 28.62 | 8.87 |
| 0.50–1.0 | | | | 24.32 | 7.54 |
| 1.0–2.0 | | | | 27.31 | 8.47 |
| 2.0–4.0 | | 35 | | | 24.15 |
| 4.0–8.0 | 45 | 65 | 69 | | 20.18 |
| 8.0–16.0 | 55 | | | | 24.67 |

As a partial replacement for cement, glass flour was used in amounts of 10, 20, and 30%. Flour is a product that is made by grinding and shredding glass into very fine particles, which gives it a light, powdery consistency but still has a relatively high density compared to some other building materials such as cement or sand. The density of the glass flour is 2.4 g/cm³. Figure 2 shows the grain size of the glass meal.

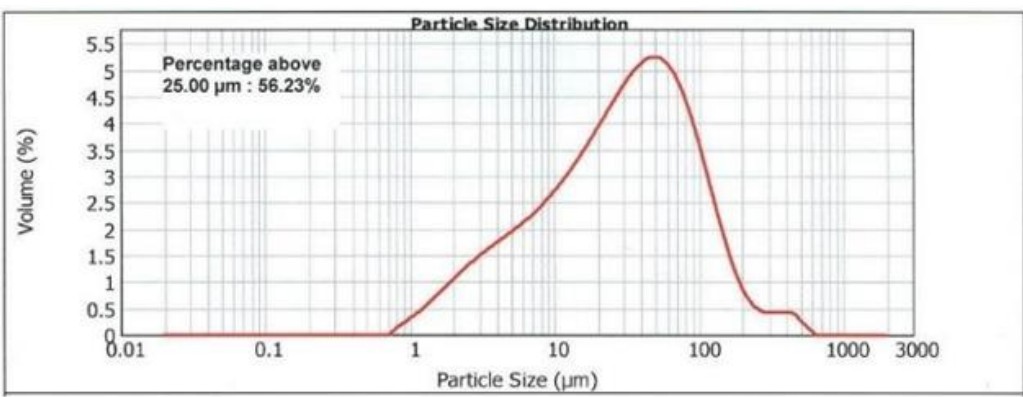

**Figure 2.** Glass flour grain size curve (manufacturer's data).

Polypropylene fibers were also used to strengthen the concrete structure. Polypropylene fibers extruded from polypropylene granulate/polymeric material ($CH_2CH(CH)_3$), bundled and cut in accordance with EN 14889-2:2006 (Bautech sp.zo.o Piaseczno near Warsaw, Poland) were also used for the tests. In the concrete mix, they form a three-dimensional supporting network resistant to gravity, thanks to which they maintain a constant level of concrete, prevent cracks, reduce plastic shrinkage, and limit the formation of shrinkage scratches [37]. Figure 3 shows the fibers and their characteristics.

**Polypropylene fibers**

- Tensile strength - 560 MPa.
- Consistency of concrete with
  4 kg/m³ of FiberMix 12 mm -
  VeBe time 7 sec.
- Fiber length - 6mm.
- Content of adhesion.
  component - 0.8%.
- Chemical and alkali resistance – excellent.
- Distribution in water - 10 sec.
- Moisture content - 1%.
- Density - 0.91 g/cm³.
- Equivalent diameter
  of single fiber - 0.02mm.

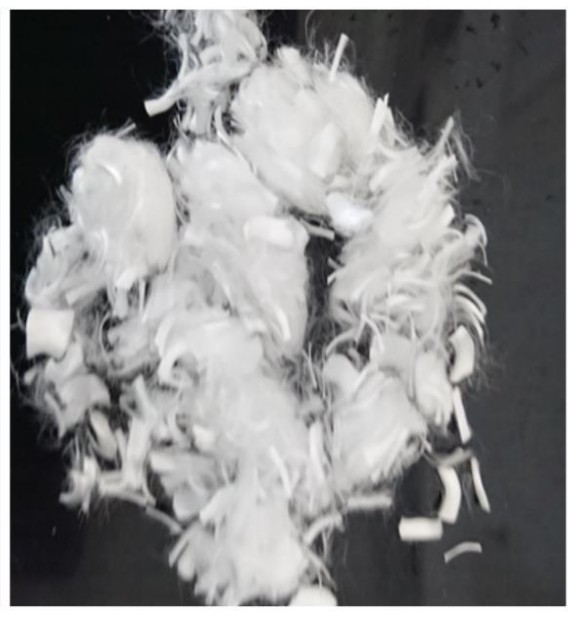

**Figure 3.** Polypropylene fibers and their characteristics.

The following types of concrete mixes have been prepared for laboratory tests:
- BZ—reference concrete without additives,
- 1M—concrete with 10% glass flour and 4% polypropylene fibers,
- 2M—concrete with 20% glass flour and 4% polypropylene fibers,
- 3M—concrete with 30% glass flour and 4% polypropylene fibers.

Table 4 shows the composition of individual concrete mixes.

**Table 4.** Concrete mix proportions by weight.

| Specification | Components of the Concrete Mix (kg/m$^3$) | | | | |
| --- | --- | --- | --- | --- | --- |
| | Water | Aggregate | | Cement | Glass Flour |
| | | Sand | Gravel | | |
| BZ concrete | 287.08 | 488.34 | 831.49 | 666.03 | - |
| Concrete with 10% M1 flour | 287.08 | 488.34 | 831.49 | 599.43 | 66.60 |
| Concrete with 20% flour M2 | 287.08 | 488.34 | 831.49 | 532.82 | 133.21 |
| Concrete with 30% M3 flour | 287.08 | 488.34 | 831.49 | 466.22 | 199.81 |

The following tests of fresh concrete mixes were carried out:

- consistency, in accordance with EN 12350-2:2019-08, reflow table method,
- density, in accordance with EN 12350-6:2019-08,
- air content, in accordance with EN 12350-7:2019-08, pressure method [38–40].

The following concrete tests were carried out:

- compressive strength, according to EN 12390-3:2019-07,
- density, in accordance with EN 12390-7:2019-08,
- frost resistance, according to 88/B-06250:2004 [41–43],
- evaluation of the effect of the addition of glass flour and polypropylene fibers on the microstructure of concrete.

The research was carried out in the Building Laboratory at the Faculty of Civil and Civil Engineering, in the Laboratory of Physical Processes at the Water Centre of the Warsaw University of Life Sciences, and in the Concrete Laboratory at the Building Research Institute in Warsaw.

The compressive strength test fc was performed after 28, 56, and 90 days of maturation in the hydraulic testing machine H011 Matest (Italy). Based on the obtained average strengths, the result was converted into cubic samples with a side of 15 cm, and the concrete class was determined.

$$f_{c,cub\ 15} = 0.95 \cdot f_{c,cub\ 10}$$

The frost resistance test was performed on samples measuring 10 × 10 × 10 cm after 28 days of concrete maturation for 150 freeze and thaw cycles. Three criteria were adopted to assess the degree of frost resistance, the fulfilment of which determines the degree of frost resistance achieved:

- no cracks on the samples after all freezing and thawing cycles,
- not exceeding the value of a 5% difference in the weight of the samples soaked in water before and after the frost resistance test,
- a decrease in compressive strength between witness and frozen samples of not more than 20%.

Three types of concrete containing polymer fibers and various amounts of glass flour (samples M1, M2, and M3) were prepared for Scanning Electron Microscope (SEM) examinations. At the first stage, from 10 × 10 × 10 cm concrete cubes, a thick slice from the middle section was cut perpendicular to the trowelling surface for each type of material. In this way, surfaces were grinded and polished. Cut and pre-polished sections of polymer composites are presented in Figure 4.

In the next step for SEM examinations, a chosen region from the pre-polished sections of each sample was cut with surface dimensions of about 20 × 20 mm. After cutting, they were dried in the oven and put into epoxy resin in a vacuum chamber. Microscopic sections were prepared in the same way as in previously published papers [44]. Figure 5 presents images of prepared samples made using an optical microscope. Before SEM examinations, samples were gold-evaporated.

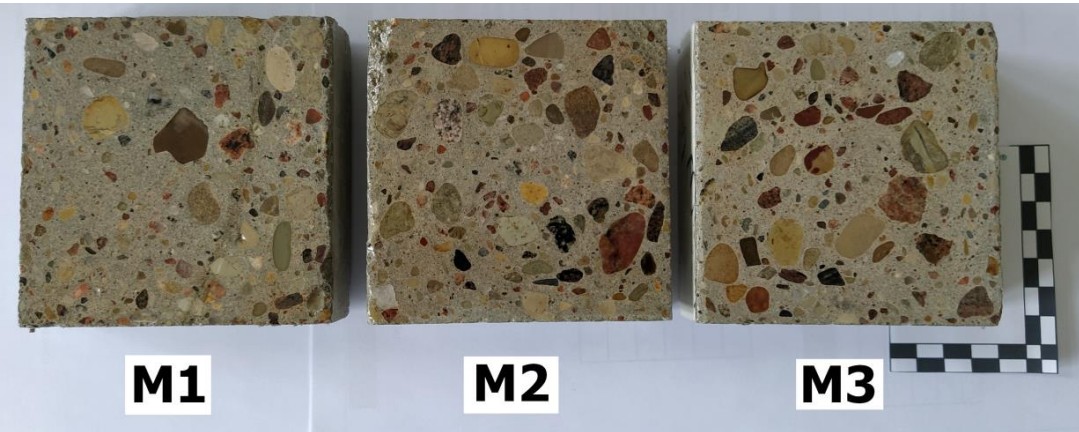

**Figure 4.** Pre-polished surfaces of concrete samples.

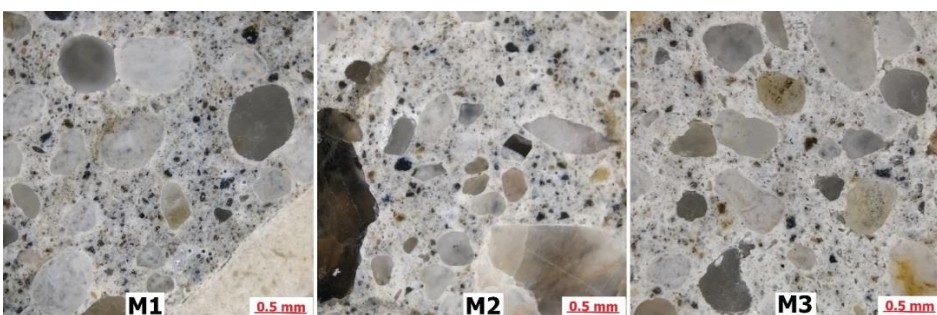

**Figure 5.** Surface of samples prepared for SEM examinations.

Microstructural analyses were carried out using SEM produced by Zeiss, model Sigma 500 VP (Carl Zeiss Microscopy GmbH, Köln, Germany). Backscattered electron (BSE) images were collected. Phase compositions and mapping were analyzed using the Energy Dispersive X-ray spectroscopy (EDX) detector produced by the Oxford model Ultim Max 40 (Oxford Instruments, High Wycombe, UK).

## 3. Results

### 3.1. Results of Concrete Mixes

Figure 6 shows the course of individual tests of concrete mixes, and Table 5 shows the results obtained.

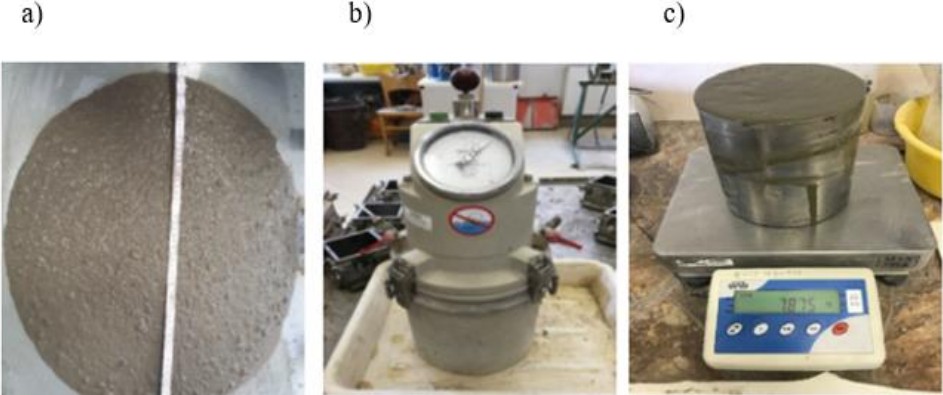

**Figure 6.** Testing of concrete mix—(**a**) consistency, (**b**) air content, (**c**) density.

**Table 5.** Test results of concrete mixes (own application).

| Specification | Consistency (mm) | Density (kg/m$^3$) | Air Content (%) |
|---|---|---|---|
| BZ | K4 | 2370 | 1.7 |
| M1 | K5 | 2249 | 2.2 |
| M2 | K5 | 2246 | 2.7 |
| M3 | K5 | 2247 | 2.9 |

Based on the results of tests carried out on concrete mixes, it was found that glass flour and fibers slightly affect its individual parameters—consistency, density, and air content. The results of the consistency tests are consistent for mixtures of M1, M2, and M3, which have obtained a consistency of K5. Glass flour is a product that does not have the ability to absorb water in a manner like traditional building materials. Glass flour is a glassy material with a smooth surface that does not absorb water in a significant way. The density of fresh concrete mix BZ obtained values of 2370 kg/m$^3$ and for mixtures with flour from 2246 to 2249 kg/m$^3$. The lowest air content was obtained for BZ samples, equal to 1.7%, while the highest air content was recorded in concrete mixes in which cement was replaced in the amount of 30%, equal to 2.9%.

### 3.2. Results of Mature Concrete

Compressive Strength

In Figure 7, the course of the compressive strength test is shown, while in Figure 8, the average test results are obtained for individual concretes made based on glass flour with polypropylene fiber.

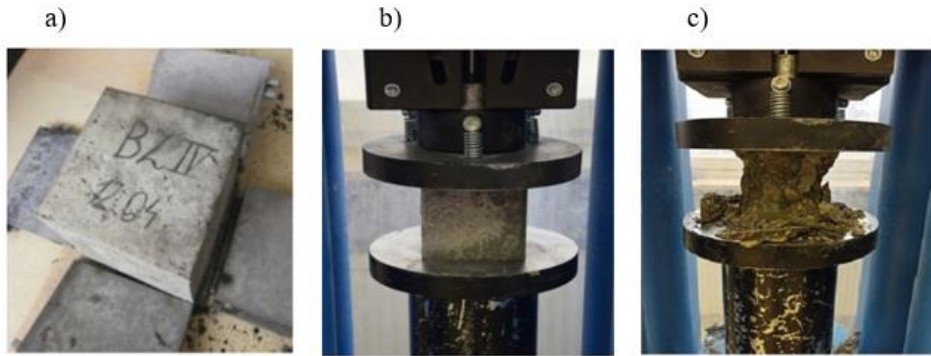

**Figure 7.** Compressive strength test (**a**) concrete sample, (**b**) sample set in the machine, (**c**) test sample.

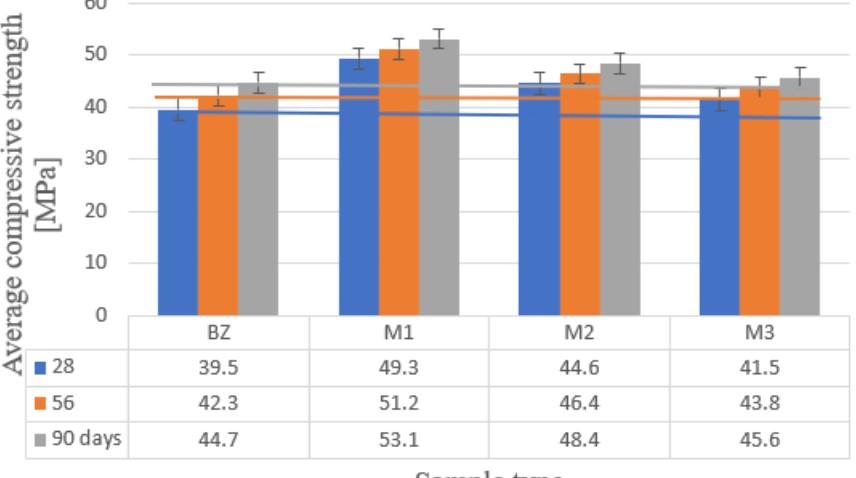

**Figure 8.** Average compressive strength of concrete.

Analyzing Figure 8, after 28 days of maturation, the highest compressive strength of 49.3 MPa was obtained by concrete in which cement was converted in the amount of 10% to flour, while the lowest strength of 39.5 MPa was achieved by reference concrete. Compared to BZ concrete, the increase was 24.8%. The highest compressive strength after 56 days, equal to 51.2 MPa and after 90 days, equal to 53.1 MPa was also obtained by concrete, in which the flour content accounted for 10%. However, the lowest compressive strength after 56 days of maturation, equal to 42.3 MPa and after 90 days, equal to 44.7 MPa, was obtained by BZ reference concrete. In relation to BZ, the increase in strength amounted to 21.0 and 18.8%, respectively. Replacing cement with glass flour with fibers increased strength over time compared to comparative concrete without an additive.

Figure 9 shows the relationship between the average compressive strength of concrete and the ratio of glass flour to cement in concrete, while Table 6 shows the obtained equations of dependencies.

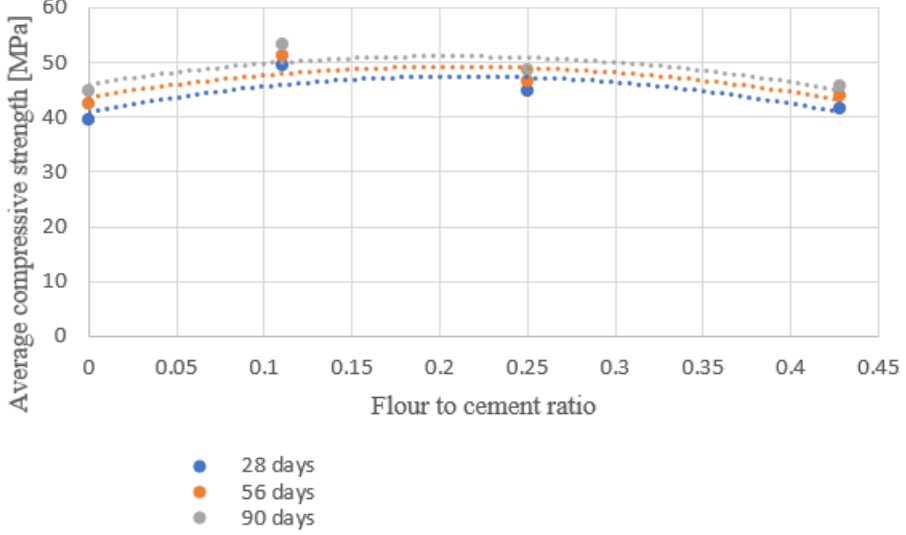

**Figure 9.** Dependence of average compressive strength on the ratio of glass flour to cement (self-support).

**Table 6.** Dependency equation for concrete with the addition of glass flour.

| Property/Days of Ripening | Dependency Equation x—Ratio of Glass Flour Content to Cement Mass | Coefficient of Determination $R^2$ |
|---|---|---|
| Compressive strength (MPa) after 28 days (M1) | $fcm = -144.1x^2 + 61.5x + 40.9$ | 0.62 |
| Compressive strength (MPa) after 56 days (M2) | $fcm = -126.3x^2 + 52.9x + 43.6$ | 0.58 |
| Compressive strength (MPa) after 90 days (M3) | $fcm = -122.4x^2 + 50.1x + 46.0$ | 0.59 |

Analyzing the graph of functions showing the dependence of the average compressive strength on the ratio of the flour content to cement, the most beneficial from the point of view of strength increase is the proportion of ash to cement in the amount of 0.20. The coefficient of determination for concrete samples ranged from 0.58 to 0.62. The obtained results indicate a low correlation between experimental and computational data.

*3.3. Density*

Figure 10 shows the results of concrete density tests.

Glass flour is a waste that has a lower density compared to cement, so using it in concrete mixes can reduce the density of concrete. The analysis of the results presented in

Figure 7 indicates that the addition of glass flour and polypropylene fibers causes a decrease in concrete density with a higher flour content. The density ranged from 2275 kg/m$^3$ for M3 concrete to 2312 kg/m$^3$ for M1 concrete. The reference concrete obtained a density of 2358 kg/m$^3$. All concrete can be classified as ordinary concrete, whose density ranges from 2000 to 2600 kg/m$^3$.

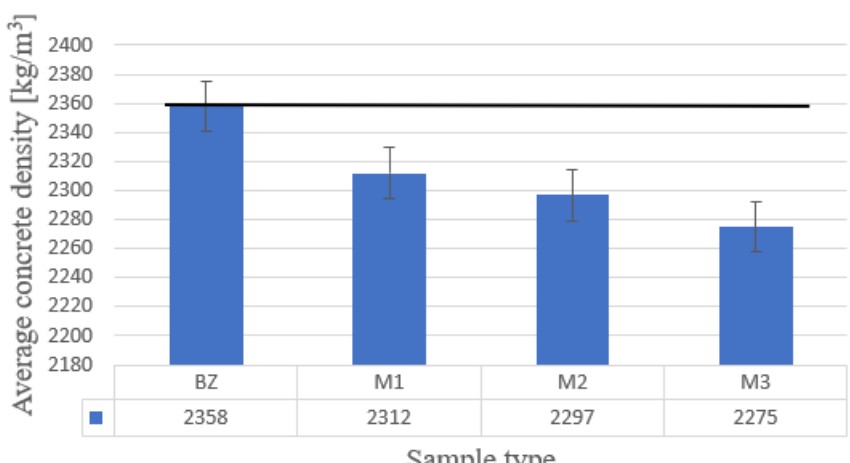

| | BZ | M1 | M2 | M3 |
|---|---|---|---|---|
| ■ | 2358 | 2312 | 2297 | 2275 |

Sample type

**Figure 10.** Average density of concrete.

### 3.4. Frost Resistance

Figure 11 shows a frost resistance test chamber and concrete samples after freezing/thawing cycles. For each type of concrete, 12 samples were prepared (6 reference samples and 6 samples for freezing/thawing cycles), of which averages were calculated (Table 7).

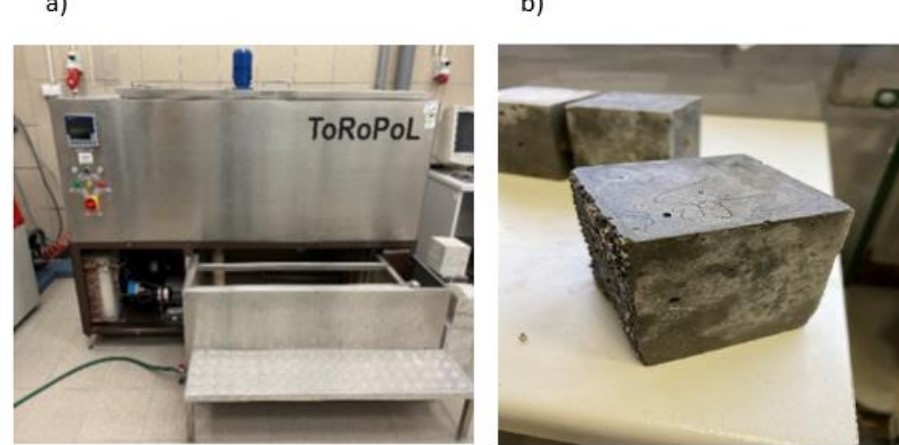

**Figure 11.** (**a**) A chamber for testing the frost resistance of concrete and (**b**) samples after the test.

The analysis of Table 7 shows that among the reference samples, the highest compressive strength of 52.6 MPa was obtained by M1 concrete, while the smallest was 43.4 MPa in BZ concrete without additives. Considering the compressive strength of samples after 150 freezing cycles, it was observed that the highest average compressive strength of 46.4 MPa was obtained by M1 concrete, in which cement was replaced with glass flour in the amount of 10%. The lowest compressive strength after 150 freezing cycles was recorded for M3 concrete. The average strength in this case was 34.2 MPa. The average decrease in the strength of the samples subjected to freezing in two cases for concrete M2 and M3 exceeded the permissible 20%. The lowest decrease in strength was for BZ concrete samples without additives. The average weight loss after freezing ranged from 0.381% for BZ samples to 0.880% for M3 samples. Samples were subjected to 150 freeze and thawing

cycles, in which cement was replaced with glass flour in the amount of 10% and concrete without additives is frost-resistant—F150. The other two types of concrete, in which cement was replaced in the amounts of 20 and 30%, turned out to be non-frost-resistant concrete. The research shows that there is a certain limitation at which the conversion of cement to glass flour causes a decrease in compressive strength after freezing/thawing cycles. In the context of these results, it can be indicated that the optimal content of the additive in the form of glass flour is 15%.

**Table 7.** Comparison of the average decrease in compressive strength of concrete samples subjected to freezing and the average loss in weight of samples made on the basis of glass flour and polypropylene fibers.

| Sample | Average Compressive Strength (MPa) | | Average Strength Drop Samples Subjected to Freezing (%) | Average Weight (g) | | Average Weight Loss (%) |
|---|---|---|---|---|---|---|
| | Reference Sample | After 150 Freezing Cycles | | Before Freezing | After 150 Freezing Cycles | |
| BZ | 43.4 | 41.2 | −5.1 | 2362 | 2353 | 0.381 |
| M1 | 52.6 | 46.4 | −11.8 | 2303 | 2293 | 0.434 |
| M2 | 47.3 | 37.7 | −20.3 | 2280 | 2265 | 0.658 |
| M3 | 44.6 | 34.2 | −23.3 | 2274 | 2254 | 0.880 |

*3.5. Microstructure of Concrete*

Figures 12 and 13 show exemplary images of the microstructure of the analyzed reference concrete without the addition of glass flour and polypropylene fibers.

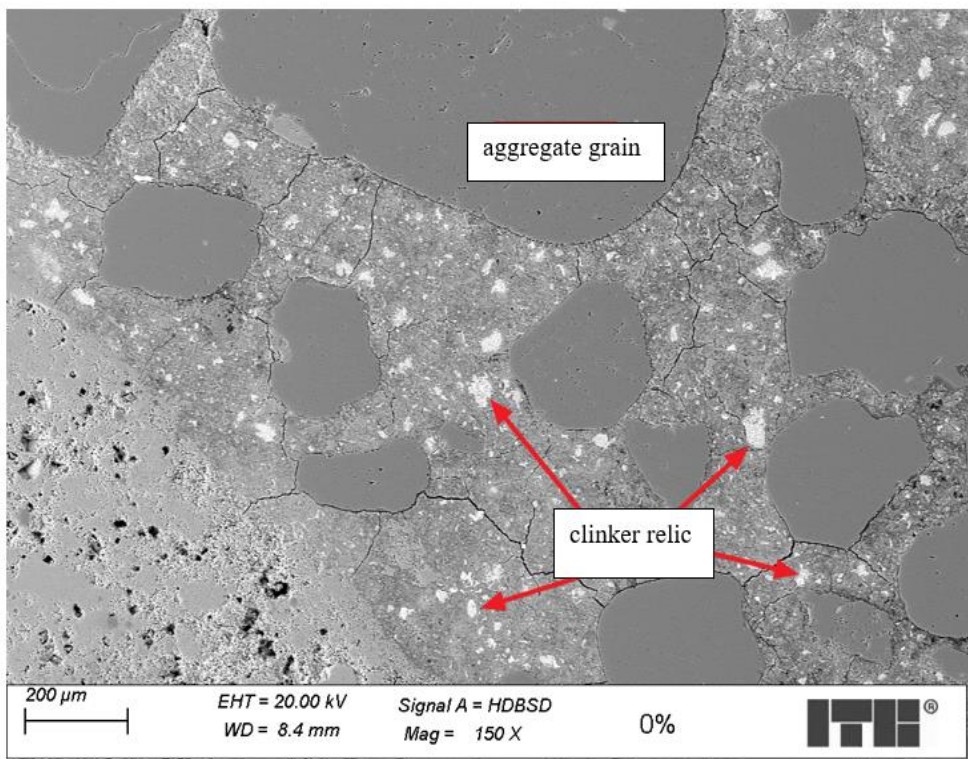

**Figure 12.** Microstructure of reference concrete (grains of clinker relics and fine quartz aggregate are marked).

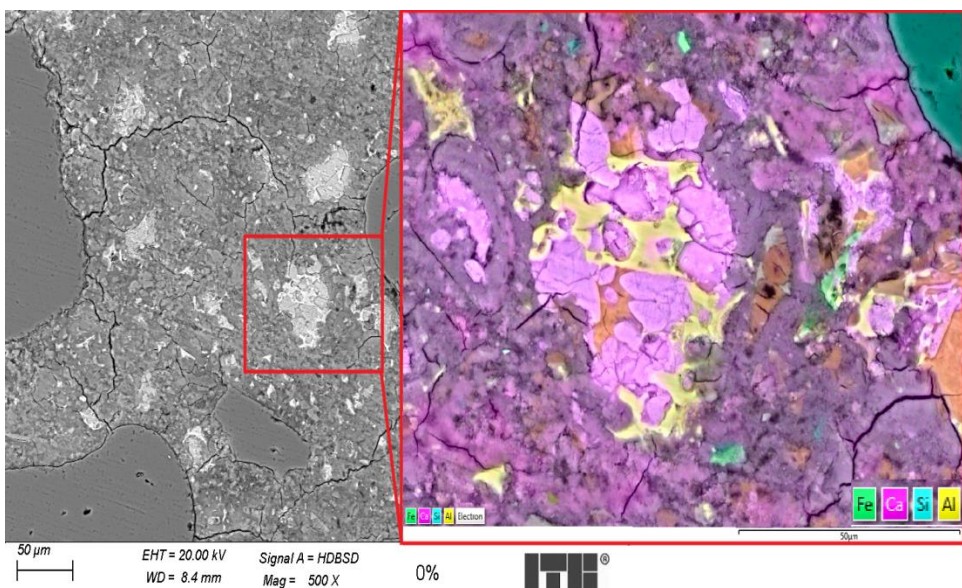

**Figure 13.** Reference concrete grout microstructure and C-S-H phase mapping around clinker relic grain.

The microstructure of the analyzed reference concrete was compact and tight, without excessive air voids or regular, spherical air pores. Numerous relics of clinker were present. No significant content of other secondary cement components was found, which confirms the use of CEM I cement for the preparation of concrete samples. The C-S-H phase was properly formed with a compact and tight structure. In most cases, the contact zone between the aggregate grains and the cement grout was tight and properly formed. Areas of discontinuous contact zone between aggregate grains and grout were few, which indicates proper mixing and compaction of the concrete mix and proper care of the internal areas of hardened concrete (Figure 13). Cracks that occurred in the cement matrix were caused by the sample preparation due to the drying of C-S-H gel.

The aim of SEM microstructural analysis was to discover the potential causes of the observed increase in compressive strength between the reference sample and the M1 sample and the causes of the observed decrease in compressive strength in samples M1, M2, and M3. Additionally, the microstructure observations might explain the slight increase in freeze–thaw resistance of concretes containing polymer fibers compared to the reference one.

Figures 14–16 present examples of the observed microstructure of concretes M1, M2 and M3.

In the cement matrix of M1–M3 samples, grains of clinker relicts were observed, along with glass flour and polymer fibers with a diameter of about 30 μm. Also, some grains of GGBS (ground granulated blast furnace slag) were observed. Some of the GBBS might be present in the CEM I cement as a minor constituent, according to the EN 197-1 standard. Comparing the microstructure of M1-M2-M3, it was observed that by increasing the content of glass flour in the cement matrix instead of a part of cement, the cement matrix became more porous and cracked. Especially when comparing the M2 and M3 samples. In the M3 sample, there were observed areas with high porosity, probably caused by the difficulties with the compaction of the samples during molding. In the M3 sample, the C-S-H gel was more porous than in other samples, which might be caused by the lower content of clinker in the cement matrix. The C-S-H phase was sealed and well developed in M1 and M2 samples, especially in areas located near the clinker relicts. Some regular spherical air voids were observed in the M1 sample.

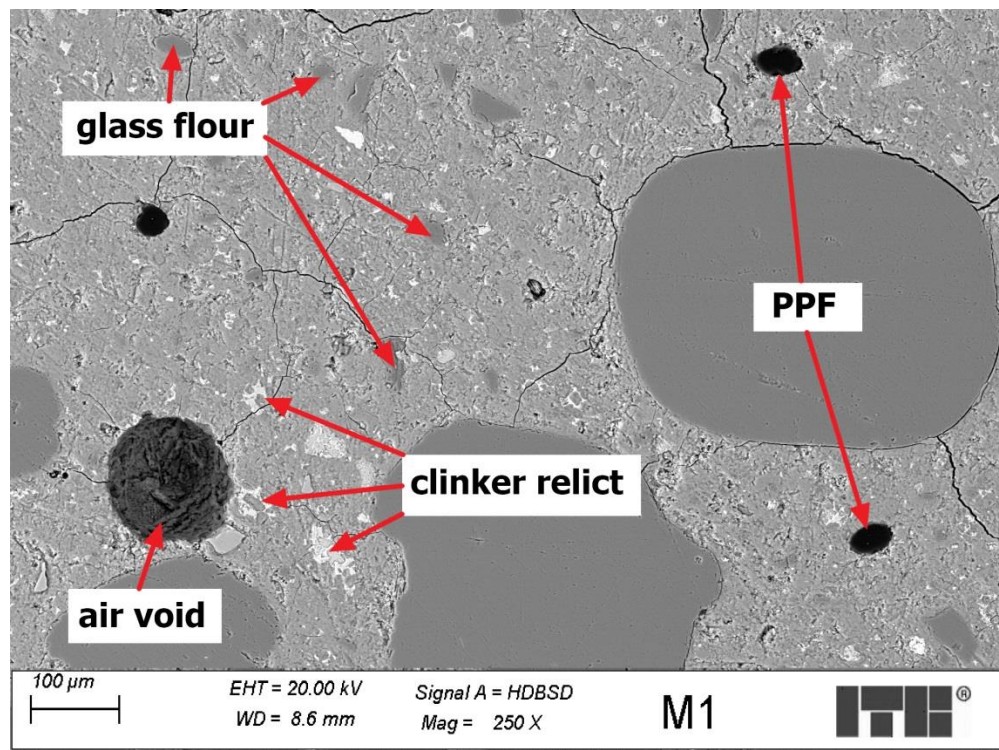

**Figure 14.** Microstructure of M1 concrete sample.

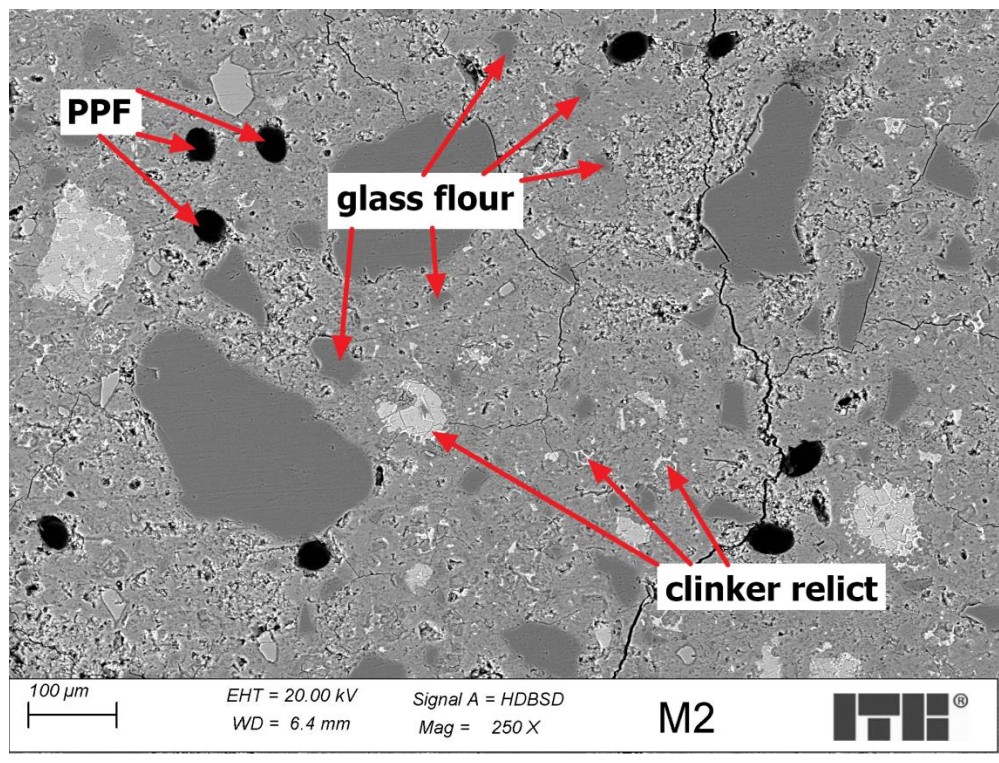

**Figure 15.** Microstructure of M2 concrete sample.

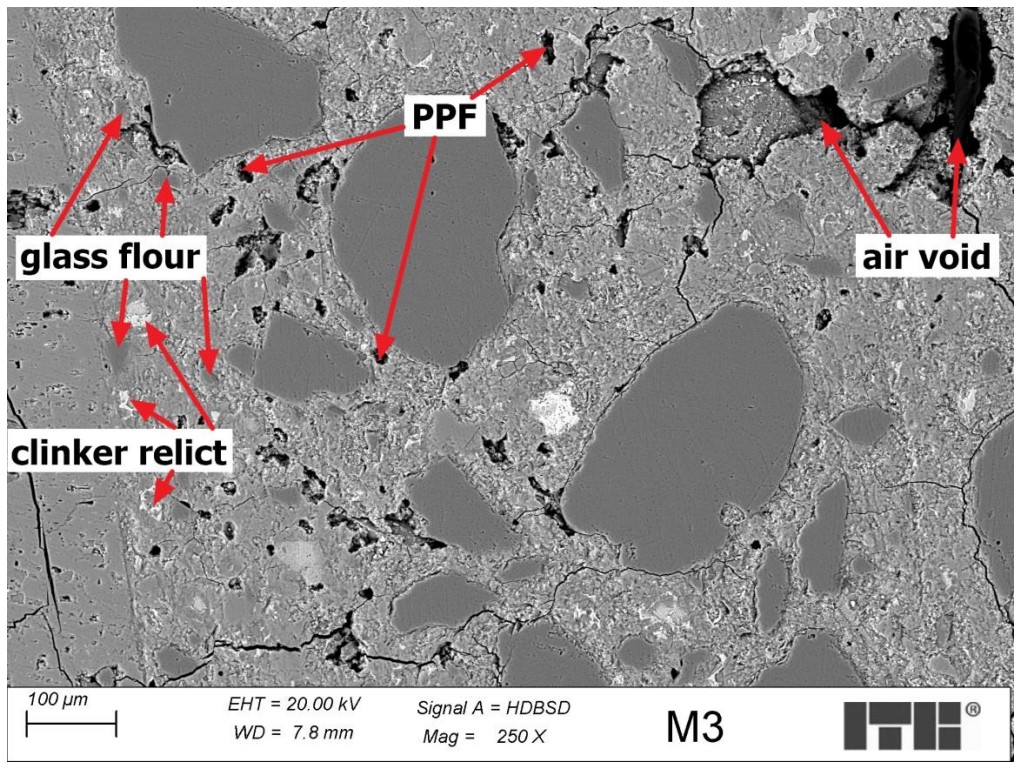

**Figure 16.** Microstructure of M3 concrete sample.

Figure 17 presents an example of the microstructure of the M2 sample and the EDX mapping of transition zone between grain of glass flour and cement matrix.

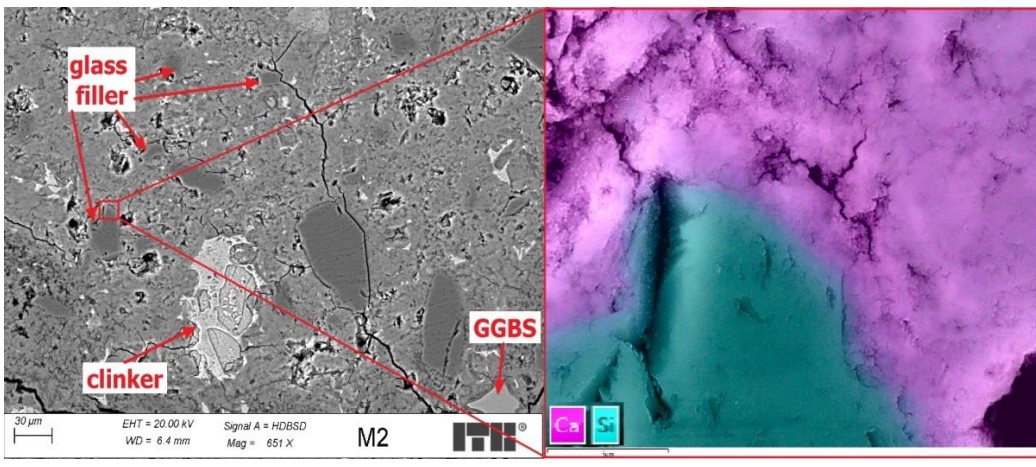

**Figure 17.** Microstructure of M2 sample and transition zone between glass flour grain and cement matrix.

The observed microstructure of the transition zone between grains of glass flour and cement matrix was mostly sealed with several cracks. The diffusion of silicon ions into the cement matrix and calcium ions in the opposite direction was observed, as shown on Figure 17 as a blue/pink halo.

## 4. Discussion

Analysis of the results of the compressive strength of concrete in different ripening periods indicates that the strength of concrete increases over time, reaching its highest values for longer periods. This phenomenon is typical for the cement hydration process, in which chemical reactions take place that cause the concrete to harden and develop its

strength. The obtained results also indicate that the content of glass flour and a constant number of polypropylene fibers have an impact on the development of strength over time. As the content of the additives used increases, differences in strength can be observed at different time stages. However, there is a limit to the optimal amount of flour to which the compressive strength increases. This suggests that the presence of additives may affect the rate and nature of the cement hydration process because of the interaction between flour, fiber, and cement, which consequently affects the development of concrete strength over time. The research confirms the results of other authors. Khatib and Negim [45] showed that cement partially replaced by glass flour has a beneficial effect on the mechanical properties of the hardened composite. Studies have shown that the maximum increase in compressive strength was achieved for samples with 10% flour, and above this value, there is already a noticeable decrease in this property [45]. Polypropylene fibers have an antispasmodic effect, protect the composite against cracks and scratches [46], and improve the frost resistance of concrete. Research with the addition of polypropylene fibers is described in Michalik and Kupisz [47], which focuses mainly on showing their impact on frost resistance. Reference concrete was compared with concrete containing class Ia (single) and class Ib (fibrillated) microfibers. The authors observed that with an increase in the percentage content of polypropylene fibers, the consistency of the concrete mix decreases, the density of the concrete mix decreases with the increase in the percentage content of single fibers, the addition of single microfibers causes a decrease in the compressive strength of the composite after 28 days, and frost resistance is significantly improved compared to reference concrete [48].

The analyzed microstructure of M1–M3 concrete samples containing additions of polymer fibers and glass flour has shown that the addition of larger amounts of glass flour as a substitute for cement might cause problems with the compaction of the concrete mix, as observed in the M3 sample. This is probably due to the decrease in the workability of concrete mix caused by the increase in water demand from glass flour.

Observed in the cement matrix of the M1 sample, several spherical air voids might be the cause of the observed slight increase in freeze/thaw resistance of this concrete. Although the amount of air voids was too low to make concrete fully resistant to this type of aggression, in the microstructures of other samples (M2 and M3), this type of air void was not observed. In the microstructure of the M3 sample, there were observed areas with high porosity and cracks, which might decrease freeze/thaw resistance even more.

Examining the microstructure of the transition zone between grains of glass flour and the cement matrix, it was discovered that silicon ions from glass migrate into the cement matrix and calcium ions in the opposite direction. It proves that grains of glass flour react with the cement matrix. It might be the cause of the observed slight increase in compressive strength of the M1 sample compared to the reference sample (without glass flour). Although the reactivity of glass flour grains might be the cause of the corrosion of cement composites by the potential alkali aggregate reaction, which needs further investigation.

## 5. Conclusions

After the tests, it can be concluded that the partial replacement of cement with glass flour and the addition of polypropylene fibers caused:

- increase in compressive strength in the tested ripening periods for samples with 10% glass flour,
- decrease in the density of the concrete mix by approximately 5%,
- deterioration of the quality of the consistency class,
- decrease in the density of hardened concrete from 2 to 4%,
- improving resistance to cyclic freezing/thawing, while obtaining frost-resistant concrete for concrete containing 10% glass flour.

Analyzing the results of microstructure observations of cement composites containing glass filler and polymer fibers, the following conclusions might be drawn:

The addition of polypropylene fibers and a small amount of glass flour causes the presence of several spherical air voids, which might slightly increase the resistance to freeze/thaw corrosion.

Adding a higher amount of glass flour might cause a problem with the workability of the concrete mix due to the increase in water demand.

Glass filler shows reactivity with the cement matrix, and in small amounts, it might cause the microstructure to seal and a slight increase in compressive strength.

Due to the reactivity of glass flour, its potential alkali aggregate reactivity needs to be investigated to prevent the potential risk of corrosion.

**Author Contributions:** Author Contributions: Conceptualization, G.R.; methodology, G.R.; software, G.R. and F.C.; validation, G.R.; formal analysis, G.R.; investigation, G.R.; resources, G.R.; data curation, G.R. and F.C.; writing—original draft preparation, G.R. and M.Ż.; writing—review and editing, G.R. and M.Ż.; visualization, E.G. and Y.T.; supervision, Y.T. All authors have read and agreed to the published version of the manuscript.

**Funding:** This research received no external funding.

**Institutional Review Board Statement:** Not applicable.

**Informed Consent Statement:** Not applicable.

**Data Availability Statement:** The data presented in this study are available on request from the corresponding author.

**Conflicts of Interest:** The authors declare no conflict of interest.

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
