# Peer review of "The Effect of Glass Flour on the Microstructure and Properties of Fiber-Reinforced Concrete: Experimental Studies"

_applsci, doi:10.3390/app132111937_

Round 1

Reviewer 1 Report

Comments and Suggestions for Authors

Comments on the Quality of English Language

Author Response

Thank you for reviewing our article and pointing out your valuable comments. All comments have been taken into account in the article.

  1. The abstract takes into account the results from the microstructure of concrete.
  2. In laboratory tests, we used glass powder, which was introduced instead of cement in the amount of 10, 20 and 30%. Thank you for pointing out the mistake.
  3. The nomenclature throughout the article has been standardized.
  4. Figure 1 is a diagram of the individual stages. The components used for the concrete mix are given, then the class of concrete and the types of samples – their designation, and finally the tests carried out on mature concrete. The drawing has been completed.
  5. There are currently no names for consistency testing in the current standard. The standard only gives the consistencies as – K1, K2, K3, K4, K5. In the text of the article, we have given the name of the consistency from the old standard in parentheses.
  6. C25/30 concrete samples are designed on the basis of EN 206+A2:2021-08 course standard. We apologize for our mistake.
  7. All the numbers in the article have been corrected. Thank you for pointing out our mistakes.

Reviewer 2 Report

Comments and Suggestions for Authors

In their study, the authors evaluate the mechanical and physical effects of glass powder and polypropylene fiber in concrete. For this purpose, mixtures with different glass powder ratios were produced. Physical and mechanical tests were carried out on concrete samples.

The following shortcomings have been identified in this manuscript;

The best results should be given in the abstract section and too much numerical data should not be given, it seems too complicated.

The introduction is very long and contains mostly information about concrete and cement. Instead, the introduction should be shorter and literature information on the use of glass powder should be increased. The following references are available for introduction.

https://doi.org/10.1016/j.cemconres.2014.01.015.

https://doi.org/10.1016/j.conbuildmat.2018.11.087

https://doi.org/10.1016/j.conbuildmat.2016.08.016.

https://doi.org/10.1016/j.jobe.2022.104849

https://doi.org/10.1016/j.conbuildmat.2020.119075.

The image of the glass flour in Figure 1 is not clear and should be corrected.

The mixing ratios of the PP ratio should be added.

The results should be compared with similar studies.

Author Response

Dear Reviewer,
      Thank You very much for Your pertinent comments on the article.

Thank you for reviewing our article and pointing out your valuable comments. All comments have been taken into account in the article. The introduction has been shortened and reference has been made to the articles indicated.

The summary includes a conclusion concerning the microstructure of concrete. Yes, we agree, the introduction was too long. The parts showing information about concrete and cement have been deleted. In the introduction, we referred to the articles indicated by the reviewer. Thank you for pointing out our mistakes.

Once again, thank You very much for Your review and we dare to ask for your favor and permission to publish the revised article.

All the Best

Authors.

Reviewer 3 Report

Comments and Suggestions for Authors The article is devoted to the study of current issues. However, there are a number of comments about the manuscript, the correction of which will ensure a better presentation of the article’s material and increase interest in it among readers.   L158-160. It must be indicated that glass flour is added to the mixture to replace a certain part of the cement.   PPF (polypropylene fiber) consumption is given in kilograms. This is incorrect, since readers do not know the total mass of the concrete mixture being prepared. The dosage of fiber is usually indicated as a percentage of the volume of the concrete mixture.   Throughout the text, the authors use two different terms “glass flour” and “glass filler”. Please indicate the difference between them or use only one of these terms.   Tab 3. There is probably an error when specifying the aggregate fraction “0.00-0.125”. The correct option is “0.00-0.0125”. In addition, the meaning of the data in the "Stage I" and "Stage II" columns should be clarified. When preparing mixtures, the fraction ratio indicated in the "Stage III" column is used. Is it so?   Fig 5. Photos look stretched vertically. Please correct this.   In the work, the reference composition was chosen incorrectly. Compositions M1-M3 differ from BZ in not one, but two parameters: quantity and presence of PPF. This does not allow to evaluate the role of glass flour on the properties of fiber-reinforced concrete, since BZ is not fiber-reinforced concrete. For a correct comparison, information on the composition of BZ+PPF is required.   Fig 9. Specify “28 days” and “56 days”.   Tab 7. It is necessary to indicate which method was used to approximate the experimental data. “Coefficient of determination R2” has low values of 0.58-0.62 (R2>0.8 is considered good). This may indicate a low correlation between experimental and calculated data. The authors should provide a justification for the adequacy of their proposed mathematical model and the practical meaning of its application. It is recommended to round the numerical values of the data in the Table to one decimal place, and the R2 values to two decimal places. In the Table cell “Dependency equation x – ratio of fly ash content to cement mass”, “fly ash” is probably mistakenly indicated instead of “glass flour”.   Fig. 11 b. The photo is compressed in height. The samples do not look cubic.   L267, L305. A link is provided to Table 6, which is not included in the text. It is necessary to add Table 6 or adjust the numbering.   L379-381. It is recommended to analyze in more detail the reasons for the increase in concrete porosity with increasing glass flour dosage. What difficulties do the authors talk about when molding samples if all compositions M1-M3 have the same consistency F5 according to tab. 5?   L417. This statement is not supported by experimental data. All samples, including BZ, have cracks.   A large number of questions are associated with the microphotographs presented by the authors. Fig. 12-17. The microphotographs clearly show cracks located in the cement stone area. The authors do not explain their presence in any way. They are probably associated with the development of significant autogenous shrinkage deformations in the cement stone, which may indicate an incorrect composition of concrete or the occurrence of some destructive processes. This question requires explanation.   It is necessary to explain how the authors identify individual components in cement stone. For example, why are the light inclusions in Fig. 12 do they refer to “clinker relic”, etc.? What is the reason for the presence of a different area with a large number of pores in Fig. 12?   Fig 17. Why do the authors consider gray areas of cement stone to be “glass filler” particles, and light gray areas to be inclusions of granulated slag (GBBS)? It is necessary to explain where GBBS could come from in the system under study (L377) since there is no mention of it in the description of the raw components. On what basis is the conclusion made about the migration of Ca and Si ions (right side of Fig. 17) between the cement stone and glass flour particles?   Figs. 14-16. It is necessary to label individual elements such as filler, PPF, glass flour, pores.

Author Response

Dear Reviewer,
      Thank You very much for Your pertinent comments on the article.

The article is devoted to the study of current issues. However, there are a number of comments about the manuscript, the correction of which will ensure a better presentation of the article's material and increase interest in it among readers.   L158-160. It must be indicated that glass flour is added to the mixture to replace a certain part of the cement.   PPF (polypropylene fiber) consumption is given in kilograms. This is incorrect, since readers do not know the total mass of the concrete mixture being prepared.

Thank you for your attention. We erroneously stated in the article that polypropylene fibers were added in kilograms. This figure is given in %.

The dosage of fiber is usually indicated as a percentage of the volume of the concrete mixture.   Throughout the text, the authors use two different terms "glass flour" and "glass filler". Please indicate the difference between them or use only one of these terms.

Thank you for pointing out this error - the naming has been standardized in the article.

Tab 3. There is probably an error when specifying the aggregate fraction "0.00-0.125". The correct option is "0.00-0.0125". In addition, the meaning of the data in the "Stage I" and "Stage II" columns should be clarified. When preparing mixtures, the fraction ratio indicated in the "Stage III" column is used. Is it so?

We agree with this remark: the aggregate fractions were incorrectly given in the table. According to the standard, the selection of aggregate for the mix by iteration should be carried out in three stages. In the first stage, 8-16 mm and 4-8 mm aggregate with the lowest sum of cavernosum and water resistance was selected, in the second stage 4-16 mm and 2-4 mm aggregate also with the lowest sum of cavernosum and water yield, in the third stage 2-16 mm and 0-2 mm aggregate.

Fig 5. Photos look stretched vertically. Please correct this.

So the photo is stretched, the tests were done on cubic samples. The photo has been corrected.

 In the work, the reference composition was chosen incorrectly. Compositions M1-M3 differ from BZ in not one, but two parameters: quantity and presence of PPF. This does not allow to evaluate the role of glass flour on the properties of fiber-reinforced concrete, since BZ is not fiber-reinforced concrete. For a correct comparison, information on the composition of BZ+PPF is required.

The aim of the work has been improved. The effect of flour and fibres on selected properties was studied.  A reference concrete mix without additives was designed for the study. A fixed amount of fibres and different powder content -10, 20 and 30% were introduced into the remaining concrete mixes.

 Fig 9. Specify "28 days" and "56 days".

The drawing has been completed.

Tab 7. It is necessary to indicate which method was used to approximate the experimental data. "Coefficient of determination R2" has low values of 0.58-0.62 (R2>0.8 is considered good). This may indicate a low correlation between experimental and calculated data. The authors should provide a justification for the adequacy of their proposed mathematical model and the practical meaning of its application. It is recommended to round the numerical values of the data in the Table to one decimal place, and the R2 values to two decimal places. In the Table cell "Dependency equation x – ratio of fly ash content to cement mass", "fly ash" is probably mistakenly indicated instead of "glass flour".

Thank you for your attention. The results were rounded according to the reviewer's direction. We agree with the reviewer that the R2 coefficient obtained small values. This shows a low correlation between experimental and computational data. In the rest of our research, we need to consider what caused such a small correlation. We are currently conducting research in which concrete mixtures containing only fibers and only flour.

Fig. 11 b. The photo is compressed in height. The samples do not look cubic.

The photo has been corrected.

L267, L305. A link is provided to Table 6, which is not included in the text. It is necessary to add Table 6 or adjust the numbering.

Thank you for this remark, the authors made a mistake in numbering the tables.

L379-381. It is recommended to analyze in more detail the reasons for the increase in concrete porosity with increasing glass flour dosage. What difficulties do the authors talk about when molding samples if all compositions M1-M3 have the same consistency F5 according to tab. 5?   L417. This statement is not supported by experimental data. All samples, including BZ, have cracks.   A large number of questions are associated with the microphotographs presented by the authors. Fig. 12-17. The microphotographs clearly show cracks located in the cement stone area. The authors do not explain their presence in any way. They are probably associated with the development of significant autogenous shrinkage deformations in the cement stone, which may indicate an incorrect composition of concrete or the occurrence of some destructive processes. This question requires explanation.   It is necessary to explain how the authors identify individual components in cement stone. For example, why are the light inclusions in Fig. 12 do they refer to "clinker relic", etc.? What is the reason for the presence of a different area with a large number of pores in Fig. 12?   Fig 17. Why do the authors consider gray areas of cement stone to be "glass filler" particles, and light gray areas to be inclusions of granulated slag (GBBS)? It is necessary to explain where GBBS could come from in the system under study (L377) since there is no mention of it in the description of the raw components. On what basis is the conclusion made about the migration of Ca and Si ions (right side of Fig. 17) between the cement stone and glass flour particles?   Figs. 14-16. It is necessary to label individual elements such as filler, PPF, glass flour, pores.

Thank you for your comments, in the article the drawings obtained from the study of the microstructure have been corrected.

Once again, thank You very much for Your review and we dare to ask for your favor and permission to publish the revised article.

All the Best

Authors.

Round 2

Reviewer 2 Report

Comments and Suggestions for Authors

The manuscript may be published in its current form.

Reviewer 3 Report

Comments and Suggestions for Authors

Dear authors, I appreciated the work you have done to improve the article. I consider it possible to recommend it for publication. Good luck in your further research!